# Uneven terrain treadmill walking in younger and older adults

Ryan J. Downey[1]*, Natalie Richer[1¤], Rohan Gupta[1], Chang Liu[1], Erika M. Pliner[1], Arkaprava Roy[2], Jungyun Hwang[3], David J. Clark[3,4], Chris J. Hass[5], Todd M. Manini[3], Rachael D. Seidler[5], Daniel P. Ferris[1]

1 Department of Biomedical Engineering, University of Florida, Gainesville, FL, United States of America, 2 Department of Biostatistics, University of Florida, Gainesville, FL, United States of America, 3 Department of Aging and Geriatric Research, University of Florida, Gainesville, FL, United States of America, 4 Brain Rehabilitation Research Center, Malcom Randall VA Medical Center, Gainesville, FL, United States of America, 5 Department of Applied Physiology and Kinesiology, University of Florida, Gainesville, FL, United States of America

¤ Current address: Department of Kinesiology and Applied Health, University of Winnipeg, Winnipeg, Manitoba, Canada

* RDowney@bme.ufl.edu

**Data Availability Statement:** All relevant data are within the paper and its Supporting Information files. Additional example scripts and data for IMU processing are will be located at the following DOI:10.5061/dryad.q2bvq83p9.

## Abstract

We developed a method for altering terrain unevenness on a treadmill to study gait kinematics. Terrain consisted of rigid polyurethane disks (12.7 cm diameter, 1.3–3.8 cm tall) which attached to the treadmill belt using hook-and-loop fasteners. Here, we tested four terrain unevenness conditions: Flat, Low, Medium, and High. The main objective was to test the hypothesis that increasing the unevenness of the terrain would result in greater gait kinematic variability. Seventeen younger adults (age 20–40 years), 25 higher-functioning older adults (age 65+ years), and 29 lower-functioning older adults (age 65+ years, Short Physical Performance Battery score < 10) participated. We customized the treadmill speed to each participant's walking ability, keeping the speed constant across all four terrain conditions. Participants completed two 3-minute walking trials per condition. Using an inertial measurement unit placed over the sacrum and pressure sensors in the shoes, we calculated the stride-to-stride variability in step duration and sacral excursion (coefficient of variation; standard deviation expressed as percentage of the mean). Participants also self-reported their perceived stability for each condition. Terrain was a significant predictor of step duration variability, which roughly doubled from Flat to High terrain for all participant groups: younger adults (Flat 4.0%, High 8.2%), higher-functioning older adults (Flat 5.0%, High 8.9%), lower-functioning older adults (Flat 7.0%, High 14.1%). Similarly, all groups exhibited significant increases in sacral excursion variability for the Medium and High uneven terrain conditions, compared to Flat. Participants were also significantly more likely to report feeling less stable walking over all three uneven terrain conditions compared to Flat. These findings support the hypothesis that altering terrain unevenness on a treadmill will increase gait kinematic variability and reduce perceived stability in younger and older adults.

**Funding:** This study was supported by grants from the National Institutes of Health (www.nih.gov). Grant U01AG061389 supported authors RJD, NR, RG, CL, EMP, AR, JH, DJC, CJH, TMM, RDS, and DPF. Grants F32AG072808 and T32AG062728 supported author EMP. The funders had no role in study design, data collection and analysis, decision to publish, or preparation of the manuscript.

**Competing interests:** The authors have declared that no competing interests exist.

## Introduction

Mobility impairments related to aging or neurological disability limit people's ability to perform activities of daily living and negatively impact quality of life. Reductions in preferred or maximum walking speed can make it difficult to cross an intersection of roads or keep up with younger family members and friends. Decreases in endurance limit the ability to move about in the community. Challenging terrain such as stairs, ramps, obstacles, or uneven surfaces can cause feelings of instability and increase the likelihood of falls.

Older adults, in particular, modify their gait in multiple ways as their mobility capabilities decline. Aging leads to weaker muscles and reduced sensory capabilities [1, 2]. Older individuals demonstrate shorter steps, slower speeds, and changes in lower limb kinematics while walking compared to younger individuals [3, 4]. Lower limb kinetic analyses also show altered lower limb joint mechanical powers and moments with aging [3, 4]. Gait stability is difficult to measure as there is no one gold standard metric, but variability measures and maximum Lyapunov exponents have strong validity [5]. Older individuals show increased kinematic variability in multiple parameters compared to younger subjects [3, 4, 6]. Increased kinematic variability has been interpreted as a loss of stability as it reflects deviations away from the limit cycle pattern and presumably a greater chance of moving outside the stable range of movement [5].

One experimental approach to identify limitations in walking ability and control among older individuals and individuals with neurological deficits is to alter the terrain. Uneven terrain is common in natural environments and is an important dimension of mobility that is often neglected in research and the clinical care of individuals with mobility impairments [7]. Walking over uneven terrain requires continuous cognitive and physical effort. The changes in height of the surface disrupt the transfer of mechanical and gravitational energy with each step, requiring more muscle work to maintain a constant average walking speed [8–10]. Compared to flat, smooth surfaces, uneven terrain also increases kinematic variability and decreases both gait stability and perceived gait stability [11–13]. Changes in foot placement, both mediolaterally and anteroposteriorly, disrupt passive dynamics mechanisms of walking [14–16] and presumably require additional control adjustments and increased muscle activation [17].

It is possible to modify treadmills to create uneven terrain for research or training purposes [8, 17–20]. Past studies have focused on comparison of a flat, smooth surface with one uneven surface and have examined healthy young adults. Uneven terrain treadmills that can be parametrically varied in their level of unevenness could be useful in identifying deficits in gait among older individuals and individuals with neurological or musculoskeletal deficits.

The overall goal of this paper was to determine how younger and older adults adjusted their walking kinematics to parametric variations in uneven terrain on a treadmill. We designed and tested a novel uneven terrain treadmill surface constructed with lightweight and easy-to-change materials. Three groups of participants (younger adults, higher-functioning older adults, and lower-functioning older adults) walked on four terrain conditions (Flat, Low, Medium, High). Our primary outcome measures were step duration variability and variability in the excursion of the sacrum. We hypothesized that there would be greater variability in the step duration and sacral excursion for the uneven terrain conditions compared to the flat terrain. We expected that the variability measures would increase with terrain unevenness. We also hypothesized that participants would report feeling less stable over uneven terrain compared to flat terrain.

## Materials and methods

This experiment was part of a larger study (*Mind in Motion*) that is still ongoing [21]. The overall goal of the *Mind in Motion* study is to determine age-related changes in the cortical

control of walking and to determine how brain activity during walking relates to clinical measures of mobility. Therefore, as part of the larger study, participants in this experiment were equipped with a wireless electroencephalography (EEG) system for the entirety of the data collection. EEG data were not analyzed in the present manuscript because it is outside the scope, but we wanted to note for transparency that participants wore an EEG setup. The EEG setup was not cumbersome for participants and did not likely impact their walking kinematics. Certain other details in the methodology described here also relate back to the design of the larger *Mind in Motion* study.

## Participant characteristics

The participants for this study were 17 healthy younger adults aged 20 to 40 years (10 females, 7 males; mean age ± SD = 23 ± 4 years) as well as 54 older adults at least 65 years of age (76 ± 6 years; 30 females, 24 males). Of the 54 older adults, 25 were higher-functioning (74 ± 4 years; 11 females, 14 males) and 29 were lower-functioning (77 ± 7 years; 19 females, 10 males). Mobility function was determined using the Short Physical Performance Battery score [22]; those with scores $< 10$ were considered lower-functioning while those with scores $\geq 10$ were considered higher-functioning. All participants were able to complete a 400 m walk test in less than 15 minutes without help from another person or a walker. Full inclusion and exclusion criteria are outlined in [21]. All participants provided informed consent before participating in the experiment, and the protocol was approved by the Institutional Review Board for the protection of human subjects at the University of Florida.

## Uneven treadmill design and customization

We modified a slat belt treadmill (PPS 70 Bari-Mill, Woodway, Waukesha, WI, USA; 70 cm x 173 cm walking surface) by adding rigid foam disks to the walking surface to create uneven terrain (Fig 1A). The lightweight yet rigid disks were made from polyurethane with a density of 8 pounds per cubic foot (128 kg/m$^3$) using a circular free-rise mold (Blockwire Manufacturing LLC, Goshen, AL, USA). We attached the disks to the treadmill using self-adhesive hook-and-loop fasteners so we could easily switch between multiple sets of uneven terrain. The loop side was attached to the treadmill surface and the hook side was attached to the disks. The spatial configuration of the disks was the same for each version of uneven terrain tested. The layout was carefully planned to avoid large gaps and adjacent disks, so the unevenness of the terrain could not be avoided. A detailed diagram is provided in S1 Fig. To accommodate the height of the disks, we removed the treadmill's incline mechanism and placed the treadmill on wooden blocks to increase the clearance with the floor.

We tested four terrain conditions which we named Flat, Low, Medium, and High (Fig 1B). For the Low, Medium, and High conditions, rigid foam disks were added to the walking surface. Each disk was 12.7 cm diameter, but the height varied by condition. The Low condition consisted entirely of 1.3 cm-high disks which were painted yellow. The Medium surface had two disk heights: 50% were 1.3 cm tall and 50% were 2.5 cm tall, all painted orange. The High surface had three unique disk heights: 20% were 1.3 cm tall, 30% were 2.5 cm tall, and 50% were 3.8 cm tall, all painted red. For the Flat condition, no disks were added to the walking surface, but green circles were painted on the treadmill belt surrounding each piece of hook-and-loop fastener which was also green. This was to ensure that the visual nature of the Flat condition was similar to the other terrain conditions. For the Medium and High conditions, which use a mixture of disk heights, the placement was consistent across participants. We achieved this by labeling the location of each type of disk by placing small pieces of color-coded duct tape next to the hook-and-loop fasteners on the walking surface and under each disk.

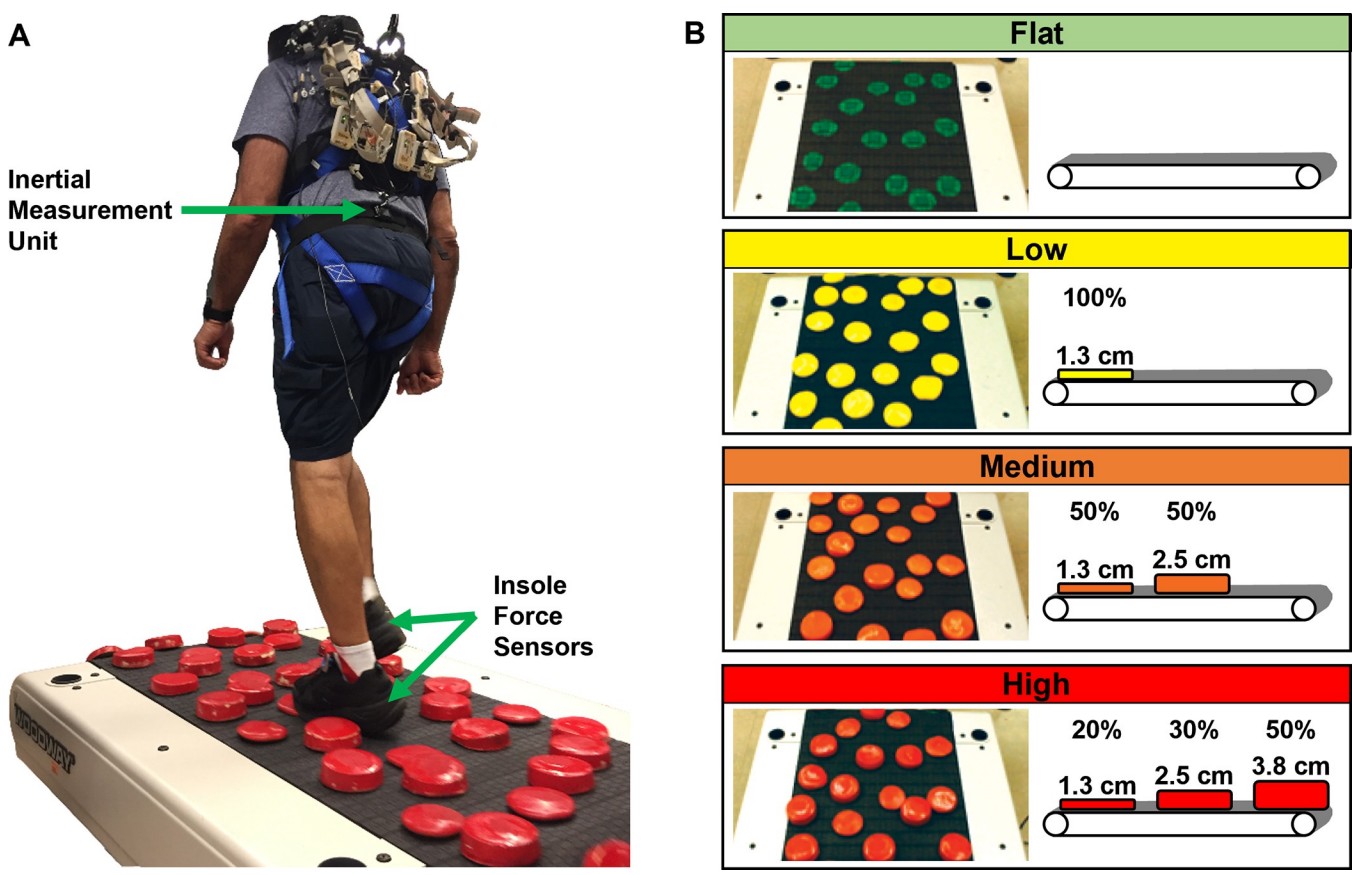

**Fig 1. Uneven terrain treadmill.** (**A**) We recorded participants' step timing along with the spatial movement of their sacrum as they walked over an uneven terrain treadmill. (**B**) We tested four terrain conditions by varying the height of the obstacles.

## Experiment protocol

**Treadmill speed calculation.** We customized the treadmill speed to each participant based on their self-selected overground walking speed. Participants walked across an overground version of the Flat, Low, Medium, and High terrain, and we used a stopwatch to record the time it took to travel the middle 3-meter portion. We calculated the overground walking speed for each terrain as the average of three trials. We then took the participant's slowest average overground speed (slowest terrain) and multiplied it by 75% to calculate a treadmill speed to target. We reduced the target speed for uneven treadmill walking because participants' self-selected walking speeds are approximately 10–15% slower for treadmill walking compared to overground walking [23, 24]. Because the speed was later kept constant across all treadmill conditions, we wanted to ensure the speed was not too fast such that participants could not complete the more difficult conditions. Participants had an opportunity to familiarize themselves with uneven treadmill walking while the treadmill speed slowly increased toward the targeted speed. We observed the participants and asked for their feedback as the speed progressively increased. We increased the speed until the participant verbally confirmed we had reached the fastest walking speed they were comfortable with or until we reached the original target speed. In either case, the participant-specific treadmill speed was kept constant across all four terrain conditions in subsequent testing. The overground self-selected walking speeds are depicted in Fig 2 for each participant group and terrain, along with the speed that was used in the treadmill trials.

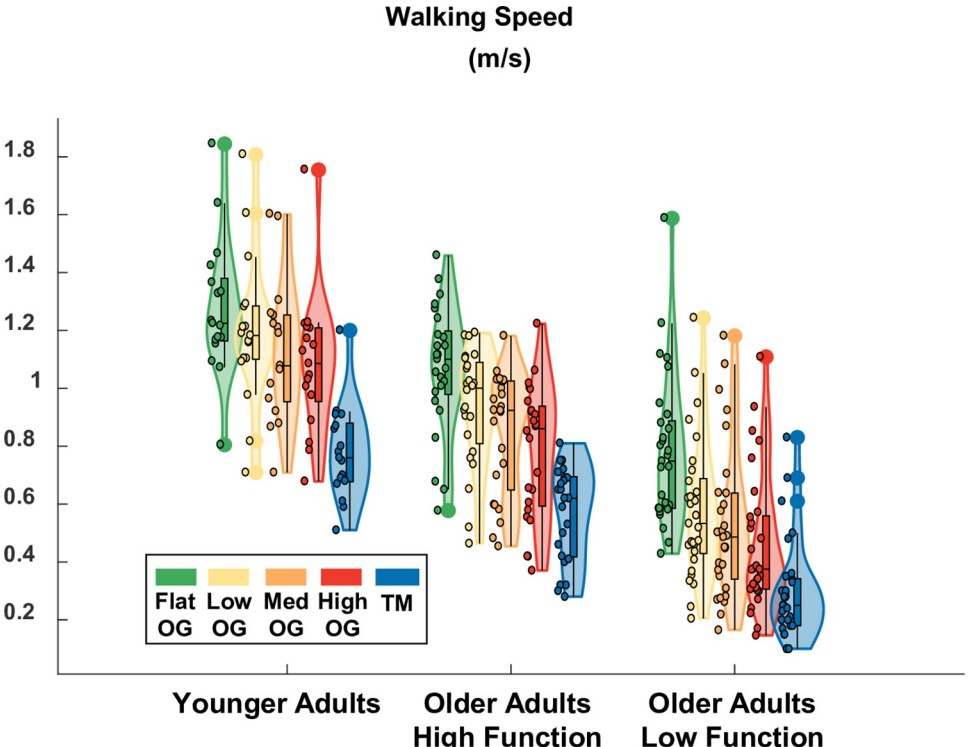

**Fig 2. Violin plot showing the distribution of the self-selected overground (OG) walking speeds and the fixed treadmill (TM) speed for each participant group.** Statistical tests were not performed, but overground walking speed tended to decrease with terrain unevenness for all groups. Lower-functioning older adults tended to walk slower than higher-functioning older adults. Younger adults tended to walk faster than higher-functioning older adults. The shaded regions represent the distribution of the data (across participants) by estimating the probability density function; each shaded region has equal area. The bottom of the box is the 25% percentile. The top of the box is the 75% percentile. The horizontal line in the middle of the box is the median. Whiskers extend from the bottom of the box to the smallest observation within 1.5 times the interquartile range. Whiskers similarly extend from the top of the box to the largest observation within 1.5 times the interquartile range. Individual data points lying outside the whiskers are plotted as large circles centered on the violin. All individual data points are plotted on the left half of each violin as small circles.

**Pseudorandomized conditions.**   Participants performed a block of two treadmill walking trials per condition. Each trial was 3-minutes long with breaks in between as needed. Conditions were pseudorandomized. Specifically, the order was selected randomly with the stipulation that the less uneven (Flat, Low) and more uneven (Medium, High) conditions be counterbalanced. For example, if the first block was Flat or Low, then the second block was Medium or High, and vice versa. This led to 8 unique orders of conditions.

**Participant safety.**   Participants wore a safety harness attached to an overhead rail (Portable Track System, Solo-Step, North Sioux City, SD, USA) to catch them in the event of a fall. Research staff members were present directly behind the treadmill and to the sides, generally out of view, to ensure that participants did not stray off the treadmill accidentally. The treadmill's handrails were removed so that participants could not hold onto them for support during walking. Participants were allowed to briefly grab onto an experimenter's arm in the immediate moment of a stumble or loss of balance, if it were to occur. However, in this scenario, they were also instructed to let go of support as quickly as possible and return to normal walking. Similarly, we allowed participants to hold onto experimenters' arms for support while the treadmill was accelerating or decelerating, but we required participants to let go and walk on their own before a trial could officially start. If a participant felt they could not walk without

support for a given condition, the condition was skipped, and data were not recorded. This only occurred for one participant, an older lower-functioning adult who did not want to perform the High condition without letting go of support.

## Data processing

**Temporal measures.**   We recorded the ground reaction force of each foot with capacitive shoe insole sensors (loadsol 1-sensor, Novel Electronics Inc., St. Paul, MN, USA) at 200 Hz. These sensors have good reliability [25] and have been previously validated against an instrumented treadmill [26]. We used custom MATLAB scripts for all data processing. We computed heel strike events for each foot as the time points when the ground reaction forces reached 20 N, similar to [27, 28]. For each gait cycle, we calculated the step duration as the time difference between heel strikes of opposing feet. We then calculated the mean and standard deviation of the step duration across all gait cycles. Our intent was to examine steady state walking, so we implemented a routine to identify and remove outlier values that were not reflective of steady state walking (e.g., shorter or longer step durations due to a brief stumble or loss of balance). Specifically, we defined outliers as individual gait cycles with step duration values lying outside the mean ± 2.5 standard deviations. We then recalculated the mean step duration and its standard deviation, with the outliers removed. Finally, we calculated the variability of the step duration as the coefficient of variation (standard deviation expressed as a percentage of the mean). The step duration and step duration variability measures were averaged across both legs within a trial and then averaged across the two trials within each terrain condition.

**Spatial measures.**   We placed an inertial measurement unit (IMU) over the sacrum using an elastic belt and sampled the data at 128 Hz (Opal, APDM Inc., Portland, OR, USA). This sensor has been shown to reliably extract gait spatiotemporal measures [29]. IMU data were offline synchronized to the insole ground reaction force sensors so the IMU data could later be analyzed with respect to each stride (gait cycle). For each walking trial, we recreated the trajectory of the sacrum using the IMU data. For each stride, we calculated the peak-to-peak excursion of the sacrum in the anteroposterior and mediolateral directions. For each direction we then calculated the variability in sacral excursion from stride to stride as the coefficient of variation. Similar to the temporal measures, we averaged the spatial variability measures across two trials within each terrain condition. We also implemented a similar outlier removal routine (removing individual gait cycles with peak-to-peak excursions lying outside the mean ± 2.5 standard deviations before calculating the coefficient of variation). Further details regarding the sacrum displacement estimation are provided in Fig 3.

**Perceived stability.**   To obtain a subjective rating of participants' perceived stability, we used a modified version of the Rate of Perceived Stability scale from [30]. The modified scale spans five unique ratings from steady to very unbalanced. After each 3-minute walking trial, we asked participants to rate the trial by asking them a series of questions. We first asked, "did it feel like work to keep your balance?" Based on their answer (Yes or No), we either assigned a stability rating or proceeded to ask the next question. The flow chart for assigning a perceived stability score is provided in Fig 4. Stability ratings were not averaged across trials within a condition since we wanted to maintain the original set of 5 unique ratings (no half ratings) for subsequent analysis (ordinal logistic regression).

## Statistical methods

We applied a linear mixed effects model to each of our four behavioral outcome measures (step duration, step duration variability, anteroposterior excursion variability, and

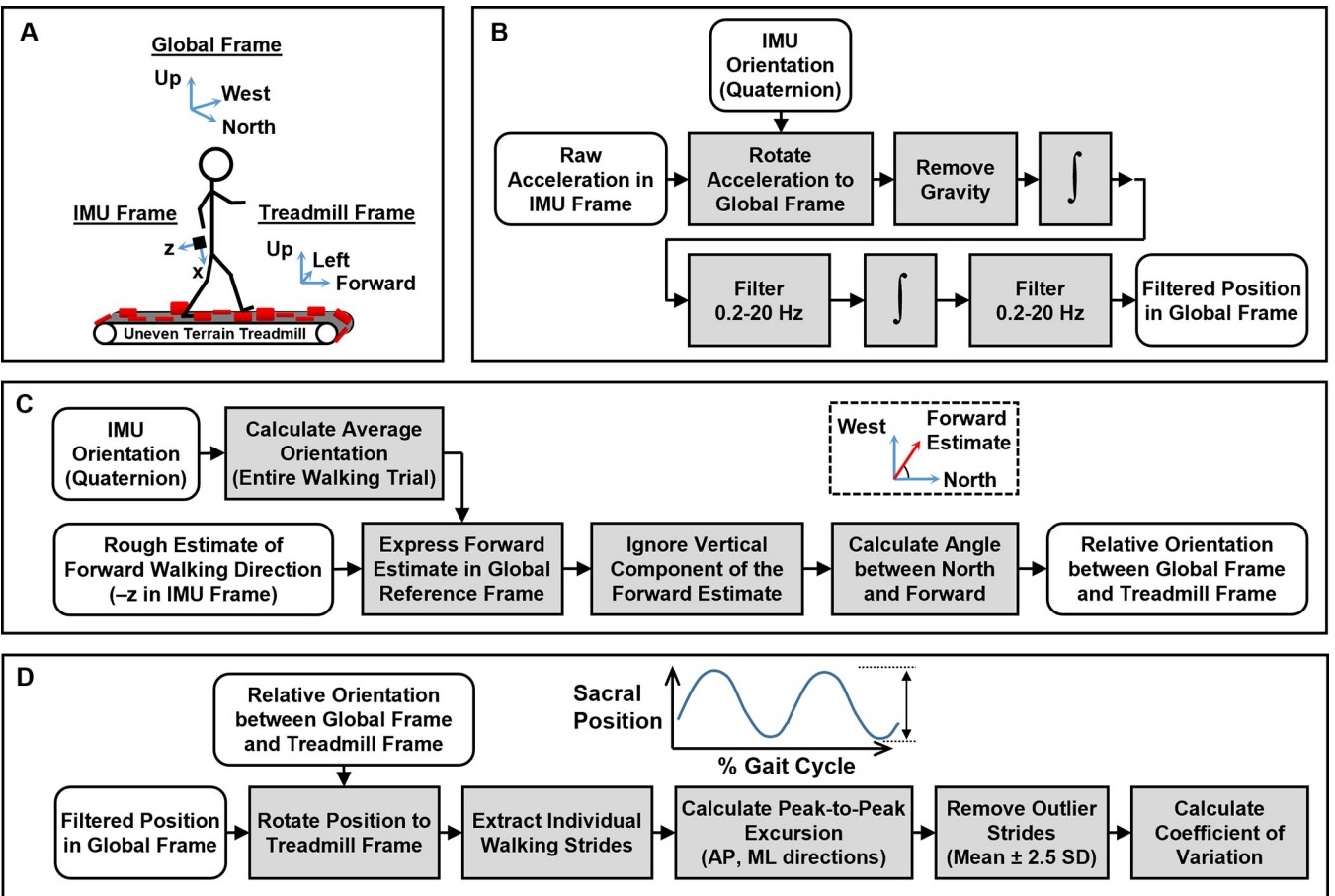

**Fig 3. Inertial measurement unit (IMU) processing pipeline.** (**A**) The IMU data, initially expressed in its local frame, was temporarily converted to a global frame, and ultimately analyzed in a treadmill fixed frame. (**B**) We calculated the movement of the sacrum in a global frame using the raw accelerations along with precalculated orientation information which was readily available on export (Motion Studio). (**C**) We used the average orientation of the IMU to find the forward walking direction. (**D**) We calculated the peak-to-peak excursion of the sacrum stride-by-side, in both the anteroposterior (AP) and mediolateral (ML) directions. Finally, we quantified the variability of the movement using the coefficient of variation.

mediolateral excursion variability) using R (carData and nlme packages). Since the treadmill speed varied across subjects, we first regressed out the common effect of treadmill speed on the raw outcome measures and subsequently fit mixed effects models on the speed-corrected measures. The first predictor for each model was the terrain (Flat, Low, Medium, High) and the second was the participant group (younger adults, higher-functioning older adults, lower-functioning older adults). We also included terms for group-terrain interactions. The baseline values for the terrain and group variables were selected as Flat and higher-functioning older adults, respectively. We accounted for participant specific inhomogeneity through random effects after adjusting for terrain effect, group effect and their interactions. We used a p-value cutoff of 0.05 for determining statistically significant effects and a cutoff of 0.1 to identify trends in the data. We estimated effects sizes using a Wald Chi-Squared test as sqrt (chisq test / sample size). Under this formula, a value of 0.1 is considered a small effect, 0.3 a medium effect, and 0.5 a large effect.

We used the coefficient of variation to quantify walking variability, so the raw values are reported as percentage points (100% × Standard Deviation / Mean). In the Results, we report the group and terrain effects in terms of the original units (percentage points); we do not take

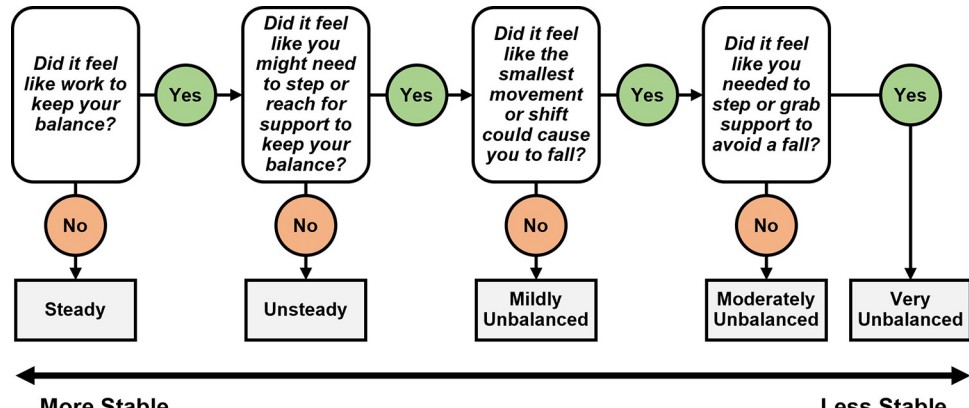

**Fig 4. Flowchart for determining a participant's perceived stability.** We asked participants to rate their perceived stability by answering a series of yes or no questions. We instructed them to only consider the official 3-minute portion of the trial when the treadmill speed was held constant. We asked them to ignore the transient moments before and after (i.e., when the treadmill is accelerating to the target speed or decelerating to a stop).

one coefficient of variation and divide it by another. For example, if the coefficient of variation increased from 4% (condition A) to 6% (condition B), we later report this effect as an increase of +2 points rather than an increase of +50%.

Data from three lower-functioning individuals were excluded from the analysis because they did not perform the protocol as intended. These participants walked at the slowest treadmill speeds of all (0.05, 0.05, 0.07 m/s). We excluded the data from these three participants because of technical issues arising from the slow treadmill walking speeds which caused these participants to frequently run out of room at the front of the treadmill and pause their walking. These pauses, which were not necessarily reflective of being challenged, were difficult to differentiate from pauses in walking due to being challenged by the terrain condition. Therefore, we limited our dataset to the 17 younger adults and 54 older adults who could walk on the uneven terrain treadmill at speeds $\geq 0.1$ m/s.

There were missing values for 2 out of 284 behavioral observations (71 participants, 4 conditions per participant). In one case, the participant was a lower-functioning older adult who was unable to perform the High condition but could complete the other uneven terrain walking conditions. In the second case, we encountered a technical issue that prevented us from recording the Low condition for a lower-functioning older adult (depleted batteries). Since the proportion of missing observations was small, the models were fit without the missing data (using 282 out of 284 observations).

For the perceived stability outcome measure, we fit an ordinal logistic regression model rather than a linear model since the values are drawn from a discrete set of ordered ratings (steady, unsteady, mildly unbalanced, etc.). Along these lines, we purposefully did not average the two trials together within a condition as that would introduce half-ratings (e.g., halfway between steady and unsteady). Instead, we kept the ratings from the individual trials separate (repeated measures). Thus, there were approximately twice as many observations for this analysis. Terrain and Participant Group were used as predictors with the baseline values set to Flat and higher-functioning older adults, respectively. Confidence intervals (95%) were constructed on the odds ratio for each model term to check for statistically significant effects. Interaction terms were not included for the perceived stability outcome measure due to issues with model convergence.

## Results

### Step duration

Terrain was not a significant predictor of step duration; effect size (ES) = 0.11. The High terrain elicited shorter step durations than Flat but only at the trend level (-.14 seconds; p-value = 0.071). Meanwhile, Group was a significant predictor of step duration (ES = 0.46), even after accounting for differences in walking speed. Compared to higher-functioning older adults, the lower-functioning group walked with longer step durations over Flat terrain (+0.37 seconds; p-value = 0.000). Meanwhile, younger adults walked with shorter step durations than higher-functioning older adults over Flat terrain (-0.26 seconds; p-value = 0.003). There were no significant interactions between Terrain and Participant Group. A table with the full statistical results is provided in S1 Table.

### Step duration variability

Terrain was a significant predictor of step duration variability (ES = 0.22), which increased as terrain unevenness increased (Fig 5) and roughly doubled in value from Flat to High for all participant groups. Compared to flat treadmill walking which had a baseline step duration variability of 5.0% in higher-functioning older adults, variability increased by 1.7, 2.2, and 3.9 points for the Low, Medium, and High terrain, respectively. This increase was significant for Medium (p-value = 0.042) and High (p-value = 0.000) terrain, but not for Low (p-value = 0.118). Group was a significant predictor of step duration variability (ES = 0.46), even after accounting for differences in walking speed. Compared to higher-functioning older adults over Flat terrain, the lower-functioning group walked with significantly greater step duration variability (+5.1 points; p-value = 0.000) and young adults walked with significantly less variability (-3.4 points; p-value = 0.005). There was only one significant interaction term in the model which was the term corresponding to High terrain and lower-functioning older adults (+3.2 points; p-value = 0.028). This term indicated that the effect of the High terrain (calculated relative to Flat) was stronger for lower-functioning older adults than for higher-functioning older adults. Thus, all groups showed a significant effect of increased step duration variability for the Medium and High terrain conditions compared to Flat, with a differential (stronger) effect in lower-functioning older adults. A table with the full statistical results is provided in S2 Table.

### Anteroposterior sacral excursion variability

Terrain was a significant predictor of anteroposterior sacral excursion variability (ES = 0.21), which increased as terrain unevenness increased (Fig 6). Compared to Flat terrain treadmill walking which had a baseline variability of 21.4% in higher-functioning older adults, variability increased by 3.7, 5.2, and 9.0 points for the Low, Medium, and High terrain, respectively. This increase was significant for Medium (p-value = 0.000) and High (p-value = 0.000) terrain but not for Low (p-value = 0.160). Group was a significant predictor of anteroposterior variability (ES = 0.64), even after accounting for differences in walking speed. Compared to the higher-functioning older adults over Flat terrain, the lower-functioning group walked with significantly increased anteroposterior variability (+16.2 points; p-value = 0.000) and younger adults walked with significantly less variability (-13.5 points; p-value = 0.000). There were no significant interaction effects. Thus, all groups showed a significant effect of increased sacral variability in the anteroposterior direction for the Medium and High terrain conditions compared to Flat. A table with the full statistical results is provided in S3 Table.

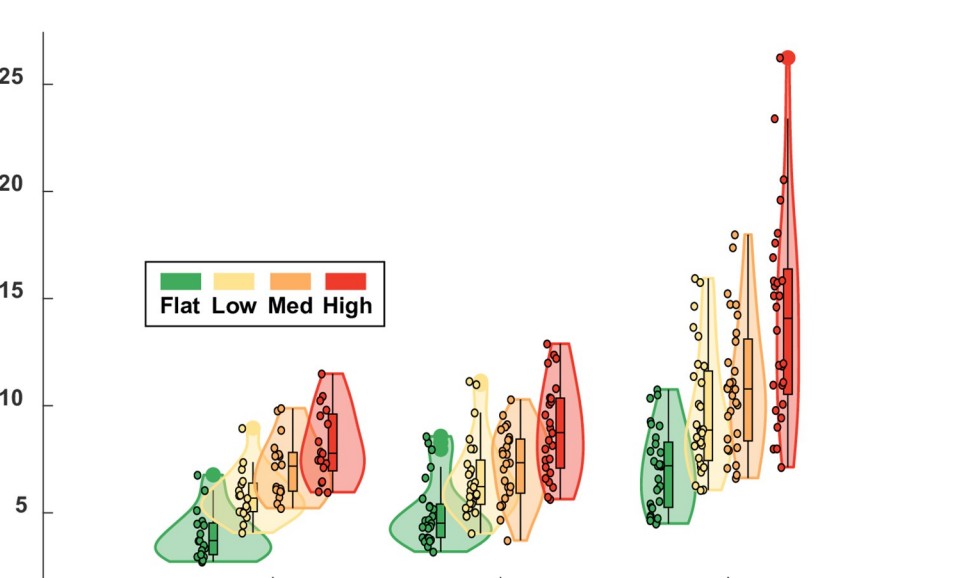

**Fig 5. Violin plot showing the distribution of the step duration variability for each terrain condition and participant group.** All groups showed significantly increased step duration variability for the Medium and High terrain conditions compared to Flat. Compared to higher-functioning older adults over Flat terrain, lower-functioning older adults walked with higher step duration variability and younger adults walked with less variability. The effect of the High terrain condition (relative to Flat) was stronger for lower-functioning older adults compared to higher-functioning older adults. The shaded regions represent the distribution of the data (across participants) by estimating the probability density function; each shaded region has equal area. The bottom of the box is the 25% percentile. The top of the box is the 75% percentile. The horizontal line in the middle of the box is the median. Whiskers extend from the bottom of the box to the smallest observation within 1.5 times the interquartile range. Whiskers similarly extend from the top of the box to the largest observation within 1.5 times the interquartile range. Individual data points lying outside the whiskers are plotted as large circles centered on the violin. All individual data points are plotted on the left half of each violin as small circles. Raw data values are plotted for readability. Statistics were performed on data that were corrected for walking speed. Removing the effect of walking speed, which differed across participants, improved the statistical estimate of Group effects.

## Mediolateral sacral excursion variability

Terrain was also a significant predictor of mediolateral excursion variability (ES = 0.27), which increased as terrain unevenness increased (Fig 7). Compared to Flat treadmill walking which had a baseline variability of 12.5% in higher-functioning older adults, variability increased by 2.8, 5.6, and 7.2 points for the Low, Medium, and High terrain, respectively. This increase was significant for Medium (p-value = 0.002) and High (p-value = 0.000) terrain, but not Low (p-value = 0.113). Group was a significant predictor of anteroposterior variability (ES = 0.26), even after accounting for differences in walking speed. Compared to the higher-functioning older adults over Flat terrain, lower-functioning older adults walked with significantly increased mediolateral variability (+5.6 points; p-value = 0.001). Younger adults walked with less mediolateral variability on Flat terrain compared to higher-functioning older adults, but it was not significant (-2.0 points, p-value = 0.304). There were no significant interaction effects. Thus, all groups showed a significant effect of increased sacral variability in the mediolateral

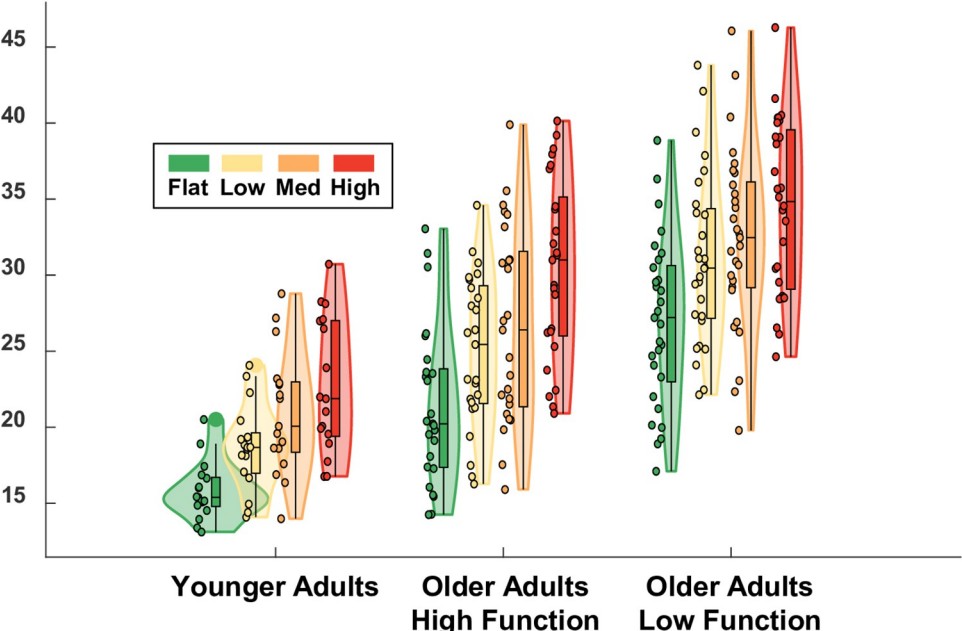

**Anteroposterior Sacral Excursion**
**Coefficient of Variation (%)**

**Fig 6. Violin plot showing the distribution of the sacral excursion variability in the anteroposterior direction for each terrain and participant group.** All groups showed a significant effect of increased sacral variability in the anteroposterior direction for the Medium and High terrain conditions compared to Flat. Compared to the higher-functioning older adults over Flat terrain, lower-functioning older adults walked with significantly more anteroposterior sacral excursion variability and younger adults walked with significantly less variability. The shaded regions represent the distribution of the data (across participants) by estimating the probability density function; each shaded region has equal area. The bottom of the box is the 25% percentile. The top of the box is the 75% percentile. The horizontal line in the middle of the box is the median. Whiskers extend from the bottom of the box to the smallest observation within 1.5 times the interquartile range. Whiskers similarly extend from the top of the box to the largest observation within 1.5 times the interquartile range. Individual data points lying outside the whiskers are plotted as large circles centered on the violin. All individual data points are plotted on the left half of each violin as small circles. Raw data values are plotted for readability. Statistics were performed on data that were corrected for walking speed. Removing the effect of walking speed, which differed across participants, improved the statistical estimate of Group effects.

direction for the Medium and High terrain conditions compared to Flat. A table with the full statistical results is provided in S4 Table.

## Perceived stability

There was a significant effect of Terrain on the perceived stability rating. Compared to Flat, all uneven terrain conditions were significantly more likely to be perceived as less stable, and the odds of a less stable rating increased with terrain unevenness (odds ratios: Low = 4.2, Medium = 17.0, High = 68.0). Lower-functioning adults were 2.8 times more likely to report feeling less stable over flat terrain than higher-functioning older adults (odds ratio = 2.8). There was no significant difference between younger adults' and higher-functioning older adults' perceived stability over flat terrain. Interaction terms unfortunately could not be included in this model due to issues with model convergence. Fig 8 displays the frequency at which each terrain condition was rated a particular stability score by each participant group. A table with the full statistical results is provided in S5 Table.

**Fig 7. Violin plot showing the distribution of sacral excursion variability in the mediolateral direction for each terrain and participant group.** All groups showed a significant effect of increased sacral variability in the mediolateral direction for the Medium and High terrain conditions compared to Flat. Compared to higher-functioning older adults, lower-functioning older adults walked with significantly greater mediolateral sacral excursion variability over Flat terrain (after accounting for differences in walking speed). Meanwhile, younger adults walked with similar mediolateral variability as higher-functioning older adults over Flat terrain (after accounting for differences in walking speed). The shaded regions represent the distribution of the data (across participants) by estimating the probability density function; each shaded region has equal area. The bottom of the box is the 25% percentile. The top of the box is the 75% percentile. The horizontal line in the middle of the box is the median. Whiskers extend from the bottom of the box to the smallest observation within 1.5 times the interquartile range. Whiskers similarly extend from the top of the box to the largest observation within 1.5 times the interquartile range. Individual data points lying outside the whiskers are plotted as large circles centered on the violin. All individual data points are plotted on the left half of each violin as small circles. Raw data values are plotted for readability. Statistics were performed on data that were corrected for walking speed. Removing the effect of walking speed, which differed across participants, improved the statistical estimate of Group effects.

## Discussion

As hypothesized, there were significant increases in step duration variability (Fig 5) and sacral excursion variability, both in anteroposterior (Fig 6) and mediolateral (Fig 7) directions, for uneven terrain compared to Flat. Increased kinematic variability during gait may indicate a loss of stability and an increased chance of falling [5]. In addition to its effect on kinematic variability, we also found there was an effect of terrain on participants' perceived stability (Fig 8). Compared to flat walking, participants were significantly more likely to report feeling less stable over uneven terrain. Altogether, these results indicate that adjusting the unevenness of the terrain challenged participants' gait.

The effect of uneven terrain was similar between the three participant groups (younger adults, higher-functioning older adults, and lower-functioning older adults). All groups demonstrated significantly increased kinematic variability over Medium and High terrain, compared to Flat, for all three of our kinematic variability measures. Additionally, only 1 of the 24

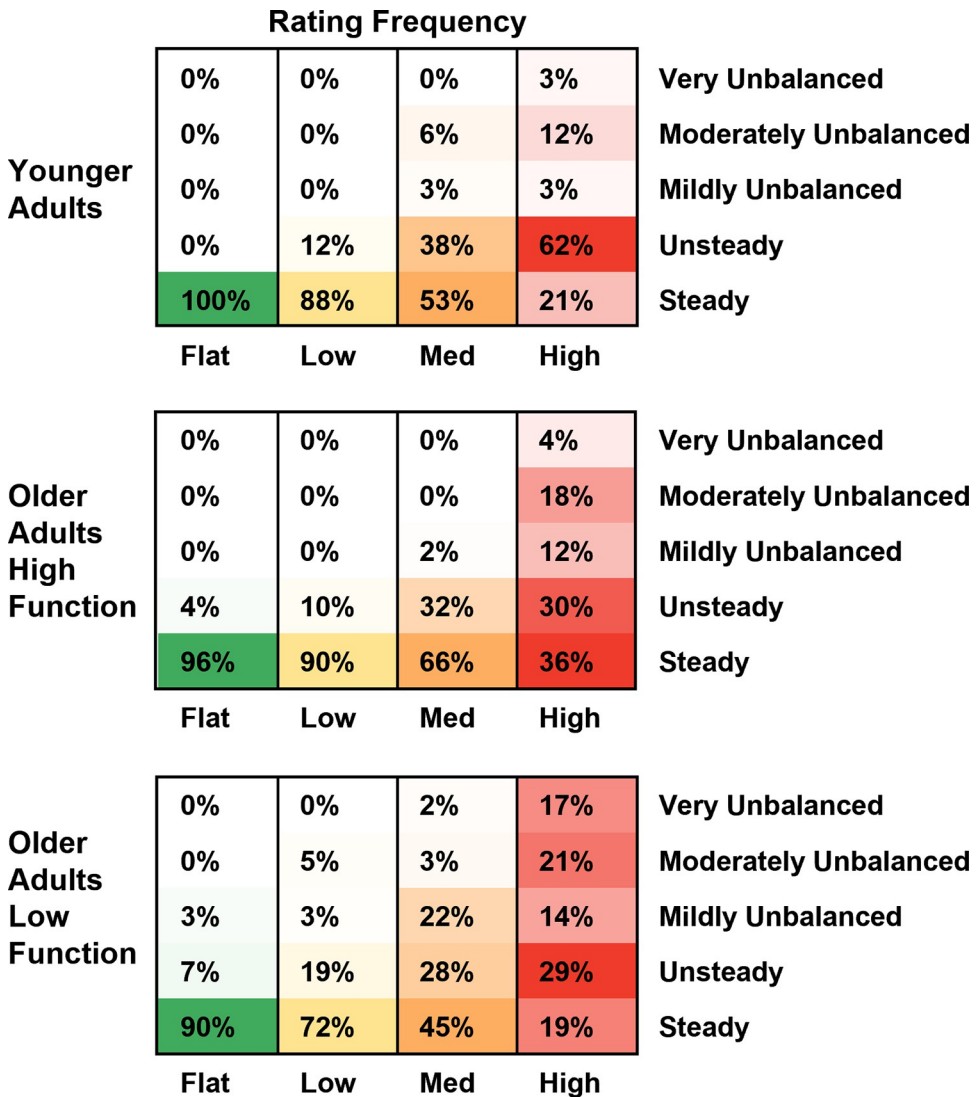

**Fig 8. Perceived stability rating frequency.** These values represent the percentage of walking trials that were rated a particular stability level as a function of the participant group and terrain. Within each group-terrain combination (i.e., within each column), colors are normalized from 0% (white) to the maximal value (solid green, yellow, orange, or red, depending on the terrain condition). This was done to emphasize the relative distribution of stability ratings and how they changed as a function of the terrain. Compared to Flat terrain, all uneven terrain conditions were significantly more likely to be perceived as less stable by all groups. Lower-functioning older adults were more likely to report feeling less stable than higher-functioning older adults over flat terrain. There was no significant difference between younger adults and higher-functioning older adults over flat terrain.

Group-Terrain interaction terms we tested was significant. In that one case, the interaction term only served to amplify the main effect of High terrain for older lower-functioning adults (increased step duration variability), rather than cancel it out. Thus, the uneven terrain treadmill surface provided a suitable challenge to stability in multiple populations, with potentially a differential effect in lower-functioning older adults.

A limited comparison can be made between our data and those from Santuz et al. [17]. In [17], the authors focused on how uneven terrain affected muscle synergies and Lyapunov exponents, abstract metrics of stability compared to the kinematic variability metrics we used (see [5] for a summary of available stability measures). Santuz et al. [17] found no difference in

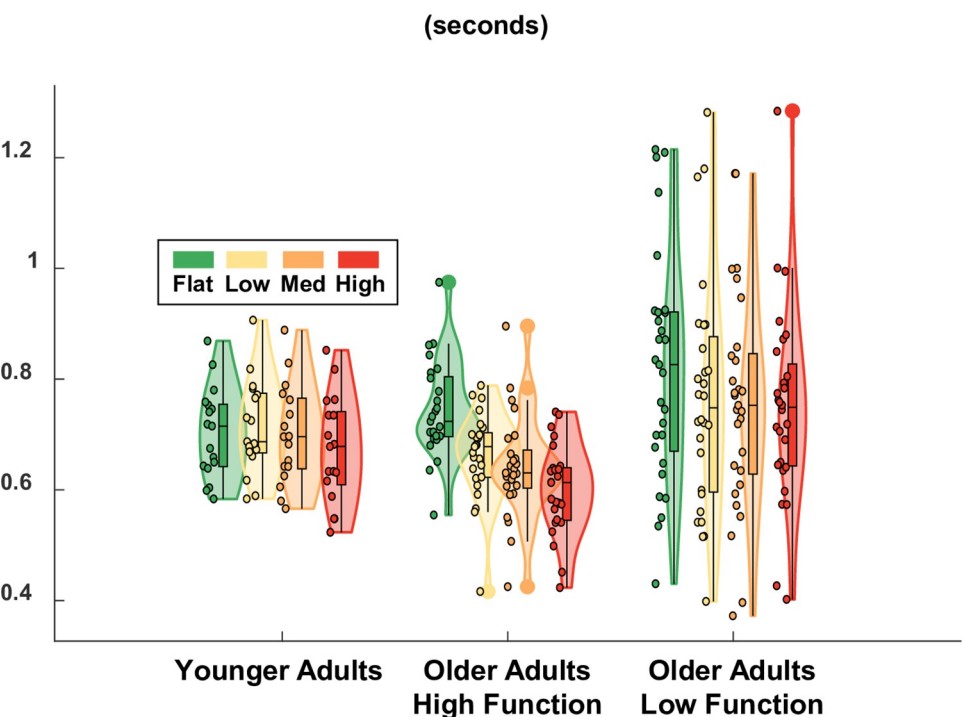

**Fig 9. Violin plot showing the distribution of the step duration for each terrain condition and participant group.**
Terrain did not significantly affect step duration, although there was a trend for shorter step durations with High
terrain compared to Flat. Compared to higher-functioning older adults on Flat terrain, lower-functioning older adults
had significantly longer step durations and younger adults had significantly shorter step durations. The shaded regions
represent the distribution of the data (across participants) by estimating the probability density function; each shaded
region has equal area. The bottom of the box is the 25% percentile. The top of the box is the 75% percentile. The
horizontal line in the middle of the box is the median. Whiskers extend from the bottom of the box to the smallest
observation within 1.5 times the interquartile range. Whiskers similarly extend from the top of the box to the largest
observation within 1.5 times the interquartile range. Individual data points lying outside the whiskers are plotted as
large circles centered on the violin. All individual data points are plotted on the left half of each violin as small circles.
Raw data values are plotted for readability. Statistics were performed on data that were corrected for walking speed.
Removing the effect of walking speed, which differed across participants, improved the statistical estimate of Group
effects.

cadence between smooth and uneven terrain in younger adults. Because the treadmill speed
was fixed, the average step duration and cadence were inversely related to each other. Thus,
their finding that cadence did not change with uneven terrain is similar to our finding that
step duration was not significantly different for uneven terrain. Note, however, that we found
a trend for the High condition to induce shorter steps. Thus, the effect of terrain on step dura-
tion may be relatively mild for treadmill walking and only evident with more challenging ter-
rain. Visually, there appeared to be an interaction between Group and Terrain, with younger
adults showing a weaker effect of terrain on step duration than higher-functioning older adults
(Fig 9). However, the interactions terms did not reach significance or trend level (p-val-
ues = 0.469, 0.437, 0.368 for Low, Medium, High).

A closer match to our study is the experiment by Voloshina et al. [8]. In [8], the average
step duration decreased from 0.57 seconds to 0.54 seconds when comparing flat to uneven ter-
rain in younger adults. In the present study, the average step duration decreased from 0.70 sec-
onds to 0.67 seconds when comparing the Flat to the High condition in younger adults. Thus,
the effect of uneven terrain on step duration in younger adults was small, but similar between

the studies (-0.03 seconds). Step durations were longer in the present study than reported in [8], for both flat terrain and uneven terrain. This was most likely due to differences in walking speed between the studies. In [8], the treadmill speed was fixed at 1 m/s for all participants. In the present study, the speed was tailored to each participant, based on their self-selected overground speed (Fig 2; average treadmill speed = 0.78 m/s for younger adults). In [8], step duration variability was originally reported as a standard deviation. Therefore, we divided this value by the mean step duration to calculate an equivalent value that would be more directly comparable to our results (coefficient of variation). In [8], step duration variability increased from approximately 2.3% to 3.3% (+1 point) whereas it increased from 4.0% to 8.2% (+4.2 points) for younger adults in the present study. This indicates that the effect of the High terrain in the present study was stronger than in [8]. This is likely due to the fact that the maximum unevenness was greater for the present study (3.8 cm versus 2.5 cm), as well as the fact that the terrain layout was less regular and therefore less predictable than in [8].

In Kent et al. [19], the authors measured the movement of the center of mass with a motion capture system as younger adults walked on flat and uneven terrain. This metric was similar to the sacral excursion metric in the present paper. Because Kent et al. [19] separately reported the mean and standard deviation of the excursion from stride to stride, rather than combining the values into a single coefficient of variation, and because exact values of the summary statistics were not provided in text, we estimated equivalent values from data in Figs 3A and 4A in [19]. Nevertheless, average values of sacral excursion variability in the two studies were comparable to each other. In the anteroposterior direction, excursion variability increased from approximately 22% to 28% (+6 points) in [19], compared with an increase from 16% to 23% (+7 points) for younger adults in the present study. In the mediolateral direction, excursion variability increased from approximately 15% to 19% (+4 points) in [19], compared with an increase from 16% to 21% (+5 points) for younger adults in the present study. Differences in baseline variability between studies could be due to differences in filtering techniques; differences in walking speed; differences in how the excursion was measured (motion capture system versus inertial measurement unit); or due to approximating equivalent values for comparison. Differences in baseline variability could also be related to differences in treadmill size. The dimensions of the walking surface in [19] were 153 cm long by 56 cm wide (TRM 731, Precor, Woodinville, WA, USA), compared with 173 cm long by 70 cm wide in the present study (PPS 70 Bari-Mill, Woodway, Waukesha, WI, USA). Thus, participants had 11% more space to move anteroposteriorly and 25% more space to move mediolaterally in the present study. Additionally, the treadmill in [19] had rails to the left and right of the participants which could further limit the range of mediolateral movement, whereas we purposefully removed them.

In addition to our behavioral results, which agreed with previous results in the literature [8, 17, 19], our study also tested a subjective measure of stability. Specifically, we employed a modified version of the Rate of Perceived Stability, which was previously used to assess the level of balance exercise intensity posed to participants [30]. Few studies have recorded subjective measures of stability during balance challenging walking tasks. Our results indicated that participants were far more likely to feel less stable over uneven terrain than flat terrain (Low = 4 times; Medium = 17 times; and High = 68 times more likely).

This was the first uneven terrain treadmill study to include multiple participant populations. Previous uneven treadmill studies [8, 17–20] included between 1–19 younger adults whereas we included 17 younger adults, 25 older higher-functioning adults, and 29 older lower-functioning adults. Our primary motivation was to determine if the uneven terrain treadmill apparatus could challenge walking, regardless of baseline walking performance. We verified that the effect of the terrain was similar across all groups, while accounting for

participant-specific and group-specific differences in baseline walking performance. We detected significant differences between the participant groups on Flat terrain, although it was not the focus of our study. After correcting for the effect of walking speed, younger adults walked with shorter step durations, less step duration variability, and less anteroposterior sacral excursion variability than higher-functioning older adults. Meanwhile, lower-functioning older adults walked with longer step durations, more step duration variability, more anteroposterior sacral excursion variability, and more mediolateral variability than higher-functioning older adults. Raw data is provided in S1 File so that other researchers can validate our findings and construct alternative statistical models to answer their own research questions.

In previous uneven terrain treadmill literature, only one uneven terrain condition was tested and compared to flat walking [8, 17, 19, 20]. Further, previous designs were not conducive to testing multiple difficulty levels, with terrain that can be removed quickly and easily. In [17], commercially available uneven terrain floor panels (Terrasensa Classic, Hübner, Kassel, Germany) were modified and attached to a treadmill. It was unclear how the terrain was attached, and the setup could be cumbersome if adapted to have multiple difficulty levels, based on the weight of the individual floor panels. The uneven treadmill in [19] is composed of many hand-shaped wooden slats with a maximum height difference of 2.2 cm. Due to the large number of pieces, replacing the surface would likely be time consuming if more conditions were developed. In [8], a secondary belt was constructed from a non-stretch canvas and secured to the treadmill, equipped with wooden blocks. The large size and weight of the setup could make it difficult to transport and store. Lastly, the treadmill in [20] had a wooden surface with a maximum height difference of 1.27 cm. No further details were given on the terrain's construction. In previous designs there was no easy way to vary the difficulty of the uneven terrain on demand. Meanwhile, the present design offers a lightweight option with the ability to quickly switch between different terrain conditions.

One potential advantage to uneven treadmill walking is that it may facilitate the investigation of mobility itself, as opposed to studying mobility under cognitive distraction (dual-task walking). In future work, this treadmill setup will be used to probe brain activity during walking [21], where multiple levels of task difficulty are needed to analyze the data within the well-supported Compensation Related Utilization of Neural Circuits Hypothesis framework [31]. Along these lines, the ability to parametrically control walking difficulty could help disambiguate dysfunctional neural processes from compensatory neural processes in future research [32]. In the present study, however, our primary objective was not to directly evaluate the gradation in difficulty between each uneven terrain, but rather to verify that each of the uneven terrain conditions challenged participants' walking compared to Flat. For this reason, we did not perform post hoc analyses to examine the pairwise differences between each uneven terrain condition. However, the present results still demonstrate a clear trend of increased kinematic variability as unevenness increases, both visually (Figs 5–7) and as evidenced by the model coefficients consistently increasing in magnitude from Low to Medium to High (S2–S4 Tables). This implies that the four terrain conditions we tested are well graded with respect to task difficulty. Thus, in future work where we examine EEG data collected during uneven treadmill walking with the same participants (as part of the larger *Mind in Motion* study), we expect to be able to extract meaningful information about the neural control of walking and how it changes with aging.

Future work could also focus on obtaining a more thorough understanding of the biomechanics of uneven terrain walking, for example, by examining other outcome measures in addition to the four presented in this study. Along these lines, we expect that other researchers will benefit from using our treadmill setup to answer various biomechanical or

neuromechanical questions. For example, this apparatus could be a useful tool to track mobility impairments and recovery. It could also potentially be used as a training paradigm to enhance recovery in the future. Our results have also demonstrated that the gradation in difficulty was similar between younger adults, higher-functioning older adults, and lower-functioning older adults, suggesting that it could be applied to a variety of populations including those with mobility impairment. Because the walking task difficulty is easily modified, either by changing the height or spacing of the disks or the speed of the treadmill, the apparatus is easily adaptable to other researchers' or clinicians' needs.

To encourage others to recreate and test our uneven terrain treadmill apparatus for their own research in the future, we chose to keep the design relatively simple without permanently modifying the treadmill surface. Attaching lightweight but rigid foam disks to the treadmill with self-adhesive hook-and-loop fasteners was a convenient and easy method to modify terrain difficulty on demand. We are providing a schematic of the spatial layout of the disks relative to the treadmill belt in S1 Fig, so that others can replicate the three uneven terrain conditions we tested. Likewise, we are providing the underlying data for the statistical analyses in S1 File, so that others can replicate our findings or repurpose the data to answer their own research questions. We are also providing example data to validate our IMU processing approach with ground-truth motion capture data, along with scripts for gait event detection, data synchronization [33].

Note that certain design choices were made which should be considered by those attempting to replicate our apparatus, as they may limit the ability to answer certain research questions. First, to allow the polyurethane disks to fit under the treadmill, we had to remove the incline mechanism and raise the treadmill on wooden blocks. Our treadmill would therefore not allow us to test the effect of these uneven terrain conditions on an inclined surface. This might be possible with other treadmills with different incline mechanisms. Second, because we designed the treadmill so the polyurethane disks could easily be removed, they would occasionally fall off if improperly attached or in the event of a foot accidentally kicking the disks from the side (mid swing) rather than the foot landing onto the disks from above. This could be viewed as an advantage, because a rigidly fixed disk is more likely to cause a trip. However, other researchers might prefer the disks to stay firmly attached during testing. In this case, more permanent solutions might be necessary. In the present study, if a piece of the terrain were to come loose during the trial, we instructed participants to ignore it and keep walking as if nothing happened. We estimate this happened a total of 39 times, spread across all 71 participants and all walking trials (0.5 pieces of terrain per participant per session). Another consideration for those interested in replicating the treadmill is that we chose to place force sensors in participants' shoes to extract gait events since the treadmill we used did not have its own force sensors. Thus, it is unclear how the uneven terrain would impact the raw measurements coming from a commercially available force-instrumented treadmill. Note that in [18], the authors were able to create a custom force-instrumented uneven terrain treadmill by placing the treadmill over two force plates. As a final consideration, note that we applied the terrain to a slat belt treadmill. This design has not been validated on a treadmill with a continuous belt.

Regarding the testing methodology, one potential limitation of this study is that we used an inertial measurement unit to recreate the trajectory of the sacrum rather than motion capture, which would have been more accurate. Additionally, due to the length constraint of a treadmill, participants have a limited amount of space (and therefore a limited amount of time) to plan their movement, compared to overground walking. This could be considered a benefit or a limitation, depending on the specific research question. Finally, another potential limitation of the present study is that we chose to visually differentiate the terrain conditions by color. It is possible that the coloring influenced participants' walking behavior (e.g., red is associated

with danger). We colored the terrain to facilitate the larger *Mind in Motion* study. Having four distinct colors helped participants recall the four terrain conditions in an imagined walking functional MRI task which is outside the scope of the present study. Future studies could replicate the present work but with different terrain coloring to test its effects.

## Conclusions

We created a novel uneven terrain treadmill design out of lightweight and easy-to-change materials (polyurethane disks and hook-and-loop fasteners). We tested four variations of terrain (Flat, Low, Medium, High), and the results showed consistent increases in kinematic variability as the terrain unevenness increased. The effect of uneven terrain was consistent across all three participant groups. These findings indicate that the uneven terrain treadmill surface was a suitable apparatus for challenging walking in multiple populations. In the future, this treadmill could be used as a tool to study other populations walking over uneven terrain. For example, it could be used to track mobility impairments and recovery after stroke; or as a therapeutic intervention; or as a testbed to evaluate the efficacy of assistive devices, such as prosthetics for amputees and exoskeletons for individuals with spinal cord injury.

## Supporting information

**S1 Fig. Obstacle positioning on treadmill.** Obstacles were placed systematically so that the terrain conditions were repeatable across participants. The total weight of the added foam obstacles was 3.5, 5.2, and 7.9 pounds (1.6, 2.4, 3.6 kg) for the Low, Medium, and High conditions, respectively.
(PDF)

**S1 Table. Statistical model results for step duration (s).**
(PDF)

**S2 Table. Statistical model results for step duration variability (%).**
(PDF)

**S3 Table. Statistical model results for anteroposterior excursion variability (%).**
(PDF)

**S4 Table. Statistical model results for mediolateral excursion variability (%).**
(PDF)

**S5 Table. Statistical model results for perceived stability ratings.**
(PDF)

**S1 File. Raw data used for statistical analyses.** Dataset includes raw values of step duration; step duration variability; anteroposterior sacral excursion variability; mediolateral sacral excursion variability; and the rating of perceived stability for each participant and for each of the four uneven terrain treadmill conditions. Also included are the age, sex, and group information of each participant; the self-selected walking speeds for overground walking for each of the four terrains; and the speeds that were used for treadmill walking. For older adults, their short physical performance battery scores are also included.
(XLSX)

## Author Contributions

**Conceptualization:** Ryan J. Downey, David J. Clark, Chris J. Hass, Todd M. Manini, Rachael D. Seidler, Daniel P. Ferris.

**Data curation:** Ryan J. Downey, Natalie Richer, Rohan Gupta, Chang Liu, Erika M. Pliner.

**Formal analysis:** Ryan J. Downey, Arkaprava Roy.

**Funding acquisition:** David J. Clark, Chris J. Hass, Todd M. Manini, Rachael D. Seidler, Daniel P. Ferris.

**Investigation:** Ryan J. Downey, Natalie Richer, Rohan Gupta, Chang Liu, Erika M. Pliner, Jungyun Hwang.

**Methodology:** Ryan J. Downey, Natalie Richer, David J. Clark, Chris J. Hass, Todd M. Manini, Rachael D. Seidler, Daniel P. Ferris.

**Project administration:** Daniel P. Ferris.

**Resources:** Daniel P. Ferris.

**Software:** Ryan J. Downey, Rohan Gupta, Chang Liu.

**Visualization:** Ryan J. Downey, Rohan Gupta, Chang Liu.

**Writing – original draft:** Ryan J. Downey, Natalie Richer, Rohan Gupta.

**Writing – review & editing:** Ryan J. Downey, Natalie Richer, Rohan Gupta, Chang Liu, Erika M. Pliner, Arkaprava Roy, Jungyun Hwang, David J. Clark, Chris J. Hass, Todd M. Manini, Rachael D. Seidler, Daniel P. Ferris.

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
