## [Decision Letter · Decision Letter 0]

23 Aug 2022

PONE-D-22-08783

Uneven terrain treadmill walking in younger and older adults

PLOS ONE

Dear Dr. Downey,

Thank you for submitting your manuscript to PLOS ONE. After careful consideration, we feel that it has merit but does not fully meet PLOS ONE’s publication criteria as it currently stands. Therefore, we invite you to submit a revised version of the manuscript that addresses the points raised during the review process.

The reviews are positive but raise a number of questions that should be addressed in a revision. In terms of the study design and analyses, you mention EEG but no data are discussed or presented, it would be good to include effects sizes and to see if more informative ways of plotting the data would be useful. There are also questions about the possible influence of treadmill speed differences and whether using average values for the statistical analysis is an appropriate choice. More details on the use and processing of the fused sensors is requested as well as a more evidence to support the arguments about mechanisms that alter gait. Please note also that PLOSONE requires the underlying data to be made available upon publication.

We look forward to receiving your revised manuscript.

Kind regards,

John Leicester Williams, Ph.D.

Academic Editor

PLOS ONE

https://journals.plos.org/plosone/s/file?id=ba62/PLOSOne_formatting_sample_title_authors_affiliations.pdf".

Reviewers' comments:

Reviewer's Responses to Questions

**Comments to the Author**

1. Is the manuscript technically sound, and do the data support the conclusions?

Reviewer #1: Yes

Reviewer #2: Yes

Reviewer #3: Yes

2. Has the statistical analysis been performed appropriately and rigorously? 

Reviewer #1: I Don't Know

Reviewer #2: Yes

Reviewer #3: Yes

3. Have the authors made all data underlying the findings in their manuscript fully available?

Reviewer #1: Yes

Reviewer #2: Yes

Reviewer #3: Yes

4. Is the manuscript presented in an intelligible fashion and written in standard English?

Reviewer #1: Yes

Reviewer #2: Yes

Reviewer #3: Yes

5. Review Comments to the Author

Reviewer #1: PLOS ONE

Submission ID: PONE-D-22-08783

Reviewer’s comments

In the submitted study, the author(s) tested young adults, high-, and lower-functioning elders in terms of gait kinematic variability when walking on four different terrains. Results revealed that the changes in terrain unevenness on a treadmill increased gait kinematic variability and reduced perceived gait stability.

There are a number of topics that need to be addressed, as presented in the following General and Specific Comments.

General Comments

• There is a mix of broader information and a lack of elaboration of research evidence concerning the mechanisms involved and the examined variables in the Introduction. It is recommended to re-organize the contents of the Introduction and to highlight the specific aim of the submitted study.

• An EEG setup is mentioned in Materials and Methods (L74-77), but no further details and data are presented. Is this correct? If yes, what was the rationale to include EEG recordings?

• Effect sizes should be reported in the respective descriptive results Tables.

• Provide additional research evidence to support the arguments about the mechanisms involved in the altering gait due to the experimental conditions in the Discussion.

Specific Comments

Abstract

• L20-25: There is a constant use of ‘We’ at the initiation of the sentences. It is recommended to rephrase.

Introduction

• See the respective General Comment.

• L39-44: It is proposed to refer to the effects of aging on gait variability since elderly and not disabled participants were tested.

• L45-49: This paragraph refers to a general content and disrupts the logic flow of the Introduction. A presentation of the examined variables could add to the context at this point.

• L53-54: Elaborate on the findings of the cited studies.

• L64-68: Elaborate on the mechanisms that alter gait because of age and uneven ground in order to build a stronger rationale for the study.

• L68-69: See the specific comment for L45-49.

Materials and Methods

• L74-77: See the respective General Comment.

• L80: Based on the demographic data of the young participants, the majority of the younger adults were in their early 20s. Was the inclusion of participants aged 40 yrs old necessary?

• L90-100: It is proposed to delete “so that others may replicate the spatial configuration we used”.

• L110-113: Was the effect of the coloring of the disks on their perception by the participants checked? In specific, as red is mainly related to danger, could it be that the participants were more cautious when negotiating the disks just by gazing the color and not by estimating the disks’ size? In addition, was the visual acuity of the participants checked? Finally, was the contrast of the colors of the disks with the color of the belt within a reasonable range?

• L125: Is there further evidence for the selection of the 75% speed?

• L170-190: Provide additional information about the familiarization process, as well as the reliability scores of the used methods.

• L173: Is there an estimate of data extraction accuracy due to the 20 N threshold?

• L176: Define the procedure to identify the outliers.

• L229-248: This part of the manuscript seems more appropriate for the data processing subsection.

Results

• State the effect sizes of the comparisons.

• Report all p-values in a similar manner, i.e., using three decimal digits.

• Not all curves are visible in Figure 4.

Discussion

• See the respective General Comment.

• L445: Cite the mentioned literature.

• L457-460: This should be mentioned as a limitation of the study.

• L491-492: Elaborate on this statement.

• L520: How often did this occur during the measurements?

• L532-533: Please clarify what is meant here.

Reviewer #2: General comments:

The authors have presented and interesting and thoughtful experiment investigating how unevenness in terrain influences gait dynamics multiple groups of participants: young adults and higher and lower functioning older adults. The design of the uneven terrain treadmill is creative and effective, and the experimental protocol and measurements are well thought out and clearly explained. The main outcome measures are clear and straightforward. Increased terrain roughness was associated with a decrease in step duration at a fixed speed.

This experiment was clearly designed for within-subjects comparisons. The treadmill speed was based upon self-selected speeds overground, which means that speed varied between subjects and there is probably a difference in average speed among the groups. Considering this, the most relevant statistical comparisons are within-subject changes across terrain conditions. Yet group effects were also analyzed and reported in the text. The authors report that lower functioning older adults tended to use longer step durations. It seems likely that this is mainly a speed effect because the treadmill speed was slower on average for this participant group. It seems completely feasible to simply include speed as an additional continuous co-variate in the statistical model, possibly with a group*speed interaction term. Such an analysis could be useful in enabling quantification of both within-subject and between-group effects.

An additional statistical point is that the justification for averaging the gait metrics before running statistics is unclear. Linear mixed effects models can handle repeated measures. Two stability ratings were maintained for statistical analysis, so it is not clear why the other data were averaged.

The authors have selected coefficient of variation as the stability metric in this study. While I agree that this is a practical and useful proxy for stability. I think it would strengthen the paper to include a discussion of the reasoning behind this, to make the rationale and validity of this choice clear to readers. (Instead of simply referencing a paper without explanation. )

Data transparency: In the figures, raincloud plots or dot plots overlaid onto box-and-whiskers would be preferable for data transparency. Violin plots add unnecessary 'ink' to graphs by duplicating my mirroring the distribution, which doesn't provide additional information. Using rainclouds (half a violin plot plus the dot scatter to the side), provides more information with similar inkspace.

Specific comments:

What was the motivation for the different colors of the uneven terrain disks?

Lines 176-177: How did you determine the outliers in the data- did you have a consistent threshold based on number of standard deviations from the mean?

Line 180-181: Why average across two trials rather than include two repeats for each subject within the statistical analysis?

Line 184: What method was used for offline synchronization of the IMU with the ground force sensors?

Fig 3: In the IMU processing pipeline, it is not clear why it would be necessary to filter both before and after integration? It seems that the correct bandpass filter before both integration steps should be sufficient. Is the 2nd filtering step getting rid of integration drift (in that case it would only need to be a low-pass filter).

Line 231: There is no theoretical limit of 100% for the coefficient of variation. The standard deviation can be larger than the mean. Consider for example a parameter that has a mean value of zero but with a distribution that spans positive and negative values. In that case, the CV would be infinity, which is one reason that this measure can be problematic for variables that have a mean value close to zero.

Line 437: Differences between the two studies here could also relate to differences in data filtering techniques.

Line 405: The comparison to Santuz and colleagues is confusing and seems incorrect as currently stated. If Santuz and colleagues found no change in cadence, then this does not seem similar to the current results. An increase in step duration with increased unevenness (as you found here) would be associated with a decrease in cadence, not a constant cadence, as found by Santuz and colleagues. If you are specifically suggesting that the most comparable conditions between the two studies is the ‘low’ unevenness condition, then this needs to be more clearly stated.

Reviewer #3: This study is about demonstrating the feasibility of a novel customizable uneven terrain treadmill by showing how some gait parameters change in three different participant groups relative to the change of the terrain unevenness. I really found the design of the treadmill promising and I also appreciate that the authors used inertial measurement units (IMU) and shoe insoles to quantify the findings rather conventional measures such as force plates and motion capture. I believe the ease of construction of the uneven treadmill and portable measures will yield different studies in gait research and rehabilitation in the future. It’s also promising to see the results of an uneven terrain on different populations so other researchers approach uneven terrain more seriously as a research, rehabilitation, and testing tool.

I have two main comments:

1) The authors did not give much detail about how they processed the IMU data to find the sacral position. Why did they use 0.2-20Hz filtering, what happens if these frequencies change? There are two integrations (Fig. 3) is there any drift due to these integrations? Did they integrate for each stride? Is there any correction applied based on constant speed walking or any gait parameter assumption such as periodicity? How did they define the rough estimate of forward walking direction? When they calculate average orientation why did they use entire trial rather than each stride? How did they define the north (magnetic north from the magnetometers of the IMU?) to calculate the angle between north and forward? How did they synchronize the shoe insoles with the IMU’s? If these details were given before in a separate paper of theirs or some else’s, it would be good to refer to those papers while answering the questions.

If the authors agree to share a sample raw IMU data (might be one trial or a part of it) and a sample script (code) that finds the sacral position, that would be nice and make the manuscript get more citations and would help to answer the questions above. Again, this is only if the authors agree to share.

Do the authors have any validity for the IMU estimates such as comparison of sacral position found from the IMU’s and motion capture? If this is done in another study, they can refer to that experiment and give its stats. Or if possible and would be much better, for 1 or 2 subjects can the authors put a motion capture marker on the sacrum and the IMU, collect some walking data on the flat treadmill (doesn’t have to be long just a couple of short trials of different speeds) and show the correlation and errors between the sacral position found from the IMU’s and motion capture? If this comparison is given, the readers trust the processing more and they would be more likely to apply this processing in their research.

2) The results overall make sense and nice, though not surprising. The authors mostly compared their results with other studies to show the validity of their novel treadmill and processing but did not comment much about what the results mean. I believe more discussion about the meaning of the results would be better. Basically, why is more variability bad or why does it mean less stability? Energetics is a dominant factor in overground human walking, it’s also dominant on uneven terrain [1,2]. On uneven terrain balance in addition to energy consumption balance might also be a dominant factor. If the sensors (vision, vestibular system, proprioceptors, cutaneous receptors so on) and the actuators (muscles) of the body have some depreciation in older adults then very likely their central nervous system is doing its best to do the task (staying on the uneven treadmill) given its depreciated sensors and actuators and this might end up being more variable. If the older adults were somehow forced to be less variable on uneven terrain the fall risk might have been bigger (just an idea). Then maybe being variable on uneven terrain is good. Is there a study that shows being variable is bad or increase fall risk on uneven terrain? I would like to see author’s comments in the discussion (maybe a short paragraph) about variability and stability in this framework. I know what I’m asking is beyond this paper and further experiments might be required (specific tasks and constraints) to answer (if possible) this question, but some comments in the discussion would be nice and would add more value to the paper.

One other thing is what would happen if the authors were to use some of the stability metrics in the reference 28? Would the terrain height and age make all or some of those metrices worse and briefly why? This is just commenting in the discussion, no analysis.

[1] https://arxiv.org/abs/2207.11224

[2] O. Darici, A. D. Kuo, Humans optimally anticipate and compensate for an uneven step during walking. eLife 11, e65402 (2022)

6. PLOS authors have the option to publish the peer review history of their article (what does this mean?). If published, this will include your full peer review and any attached files.

Reviewer #1: No

Reviewer #2: No

Reviewer #3: No

---

## [Author Response · Author response to Decision Letter 0]

22 Oct 2022

Please see attached document with point-by-point response to the Reviewers and Editor.

---

## [Decision Letter · Decision Letter 1]

22 Nov 2022

Uneven terrain treadmill walking in younger and older adults

PONE-D-22-08783R1

Dear Dr. Downey,

We’re pleased to inform you that your manuscript has been judged scientifically suitable for publication and will be formally accepted for publication once it meets all outstanding technical requirements.

Kind regards,

John Leicester Williams, Ph.D.

Academic Editor

PLOS ONE

Additional Editor Comments (optional):

Reviewers' comments:

Reviewer's Responses to Questions

**Comments to the Author**

1. If the authors have adequately addressed your comments raised in a previous round of review and you feel that this manuscript is now acceptable for publication, you may indicate that here to bypass the “Comments to the Author” section, enter your conflict of interest statement in the “Confidential to Editor” section, and submit your "Accept" recommendation.

Reviewer #1: All comments have been addressed

Reviewer #2: All comments have been addressed

Reviewer #3: All comments have been addressed

2. Is the manuscript technically sound, and do the data support the conclusions?

Reviewer #1: Yes

Reviewer #2: Yes

Reviewer #3: Yes

3. Has the statistical analysis been performed appropriately and rigorously? 

Reviewer #1: Yes

Reviewer #2: Yes

Reviewer #3: Yes

4. Have the authors made all data underlying the findings in their manuscript fully available?

Reviewer #1: Yes

Reviewer #2: Yes

Reviewer #3: Yes

5. Is the manuscript presented in an intelligible fashion and written in standard English?

Reviewer #1: Yes

Reviewer #2: Yes

Reviewer #3: Yes

6. Review Comments to the Author

Reviewer #1: In the re-submitted version, the authors addressed all the topics noted in the initial round of review thoroughly and provided adequate responses. There are no further objections.

Reviewer #2: Overall the authors have dealt satisfactorily with the concerns raised from the previous version.

Reviewer #3: (No Response)

7. PLOS authors have the option to publish the peer review history of their article (what does this mean?). If published, this will include your full peer review and any attached files.

Reviewer #1: No

Reviewer #2: No

Reviewer #3: No

---

## [Editor Report · Acceptance letter]

5 Dec 2022

PONE-D-22-08783R1 

Uneven terrain treadmill walking in younger and older adults 

Dear Dr. Downey:

I'm pleased to inform you that your manuscript has been deemed suitable for publication in PLOS ONE. Congratulations! Your manuscript is now with our production department. 

Kind regards, 

on behalf of

Dr. John Leicester Williams 

Academic Editor

PLOS ONE